# Empowering the Voiceless: Securing the Participation of Marginalised Groups in Climate Change Governance in South Africa

**Nomfundo Sibiya** [1,*] **, Mikateko Sithole** [2] **, Lindelani Mudau** [2] **and Mulala Danny Simatele** [1,3]

---

[1] School of Geography, Archaeology and Environmental Studies, Faculty of Science,
University of the Witwatersrand, Johannesburg 2000, South Africa; Mulala.Simatele@wits.ac.za
[2] Department of Forestry, Fisheries and the Environment, Pretoria 0001, South Africa;
MFSithole@dffe.gov.za (M.S.); LMudau2@dffe.gov.za (L.M.)
[3] The Global Change Institute, University of the Witwatersrand, Johannesburg 2000, South Africa
[*] Correspondence: 573380@students.wits.ac.za or npsibiya@gmail.com

**Abstract:** For many of the world's poor people, adaptation to climate change is not a choice but a reality. Existing evidence suggests that the poor, particularly those in the developing world, are the most vulnerable to any changes in climate variability and change. Using research methods inspired by the tradition of participatory research, we explore and discuss community perceptions on climate change adaptation governance in South Africa. We examine the myriad ways in which climate change adaptation policies and strategies are developed, and we systematically discuss the factors which either facilitate or hamper the involvement of all stakeholders in the development of these intervention measures. Our findings indicate that women seem to be the group of people who are mostly unaware of community initiatives, policies, and strategies for the adaptation to climate change. Thus, it is argued that, although South Africa has developed good climate change initiatives, policies, and strategies, the implementation of these policies seems to present difficulties, as those for whom they have been developed do not seem to have any knowledge of their effectiveness in helping them build resilience against extreme weather events. This study recommends that, in order to achieve successful public participation in climate change adaptation policy development, there must be an all-inclusive system which incorporates all stakeholders, including vulnerable groups.

**Keywords:** climate change governance; climate change adaptation; community participation; South Africa

## 1. Introduction

The impacts of climate change on socio-economic and environmental aspects have now become a global concern [1]. Climate change impacts have significant implications on exacerbating existing inequalities, and they are driving those with poor adaptive capacity into deeper conditions of vulnerability to shocks and stresses [2,3]. It has indeed become a social justice issue, and the late recognition of this state of affairs is indicative of both global and national failures to formulate and implement adequate adaptation policies and strategies which build the resilience of those most affected by it [4].

It is no exaggeration to argue that the most vulnerable groups to climate change lack a voice to influence and participate in any policy-making processes [5]. Their lived experiences and the challenges associated with extreme weather events are often unconsidered and unrepresented in many climate change intervention measures implemented by governments [6–8]. This exacerbates their position of vulnerability, in as much as they are the groups in society who have done the least to trigger changes in climatic conditions [9–14]. Thus, paying attention to how adaptation policies and strategies are developed, by way of considering who influences the nature of these intervention measures and who participates



in their formation, is a moral grounding issue to climate change governance [15]. This paper evaluates how climate change policies and strategies are developed in South Africa. At the core is the need to understand how climate change impacts and intersects with people's vulnerabilities and whether existing adaptation policies and strategies reflect the needs of those most impacted by it [16].

While there is a rich presence of literature focusing on rights-based analysis and emphasising a human rights approach to adaptation policy development, the pathways to comprehensive and/or transformative participation largely remain under-researched and misunderstood [17]. There is, therefore, an urgent need to move away from the instrumentalist approaches of participation and towards a system which aids in identifying conduits for realising participatory rights for different groups of the voiceless, highlighting opportunities and exposing barriers to participation at multiple scales. Such a process would bring adaptation decision-making closer to those most negatively impacted by climate change [18–21]. The current, centralized structures of governance and decision-making, which to be honest, were created and designed to deal with other social and economic challenges, and which must now address climate changes issues, need to be reformed, aligned, and tailored to deal with contemporary, climate-induced problems in specific contexts [5]. The one-size-fits-all governance approach cannot be applied to understanding the impacts of climate change on different groups of people and in diverse geographical locations [22].

Additionally, this paper investigates the question of whose voices matter in the development of climate change policies and strategies in South Africa. Thus, in this paper, we argue that vulnerable groups of people, who oftentimes lack the voice and power to influence policy direction, expend tremendous energy and vitality to change their vulnerability to climate change. The missing link, however, is the absence of governance systems and structures which are capable of harnessing the ingenuity and energy of the poor to foster meaningful and transformative public–public and private–public partnerships aimed at the inclusive participation in designing, developing, and co-implementing transgressive adaptation strategies [23]. If this gap can be bridged, it would enable the formulation of governance systems and approaches which would contribute to developing a pro-poor climate change agenda and framework for building the adaptive capacity and resilience of groups of people who are most vulnerable to climate-induced, extreme weather events and variability.

## 2. Climate Change Governance: A Synthesis of Literature from the Global to the Local Context

Climate change is a cross-cutting issue that affects all sub-sectors of an economy and presents different challenges to social, economic, and environmental systems [24]. It has been conclusively argued that, although climate change is a global concern, its impacts are disproportionally felt by the poor people living in the poor countries of the world [25,26]. Sub-Saharan Africa has been identified and is projected to be a region that will and is most susceptible and vulnerable to climate change impacts due, in part, to a large presence of poor people on the continent, weak institutional and policy frameworks, a lack of infrastructure, stagnation, deterioration in the economic systems, and a general lack of political will [27,28]. Furthermore, Africa suffers from low levels of technological advancement, education, rapid population growth, high rates of poverty, and a lack of social safety nets [18–21]. These biophysical, political, and socioeconomic stresses interact to heighten the region's susceptibility to climate change, despite emitting the lowest levels of greenhouse gases in the world [25,29–32].

Adger et al. [6] argue that climate change does not only disturb the livelihoods of poor people, but it also compromises the economic growth and national development of countries, particularly those in the developing south. A study by van der Bank and Karsten [24], observes that in countries with poor and weak economic systems and policy frameworks, the impacts of climate change always have far-and-wide-ranging effects, and

these tend to affect and subject the poor to unprecedented levels of risk to environmental hazards. Pielke et al. [33] further observe that recognising the inevitable impacts of climate change on poor people and the economies of poor countries is a vital first step in identifying and developing appropriate and strategic policy frameworks for supporting climate change mitigation and adaptation initiatives. Edifying these observations, Kurukulasuriya and Mandelsohn [34], as well as Smit and Skinner [35], advocate for pro-poor climate change adaptation policies and strategies that are based on the realities that confront poor people, and which have strong scientific bases. Furthermore, Challinor [36], supported by Niang et al. [37], is of the view that any pro-poor adaptation initiatives must support the livelihoods of those who are most vulnerable to climate change by finding effective strategies to cope with environmental changes by addressing the local and institutional barriers to the use of these strategies. In the same vein, Averchenkova [5] posits that the capacity of civil society to engage in climate change and to influence decisions seems to be generally weak in developing countries, particularly for the rural poor communities, as they have both limited resources and a limited capacity to engage. Thus, an issue-based perspective to adaptation must be the focal point in searching for an alternative perspective to the often-common bottom-up and top-down approaches, which are always the two dominant paradigms in adaptation discourses.

While it is important to consider these approaches, it is also imperative to cogitate the need for community mobilisation and participation in adaptation policy discussions, as pro-poor adaptation initiatives require that the myriad spatial contexts in which the poor live are taken into consideration [38,39]. This assertion, therefore, speaks to the importance of those mandated to formulate adaptation policies and strategies to recognize the ability of the poor to adapt to the impacts of climate change without pre-conceived assumptions. Reid and Simatele [40], for example, argue that national and local government institutions and authorities must recognize the important role that community engagement and mobilisation play in developing a comprehensive model for climate change adaptation. Cheru [41], on the other hand, argues that institutional and policy bureaucracy oftentimes tend to hamper the formulation of progressive policies to change the misery in which the poor live. Ribot et al. [42] elucidate this observation by arguing that institutional and policy frameworks can fundamentally constrain the capacity of the poor to adapt to external and internal stressors because institutions control the distribution of resources, which are the basis of any adaptation initiatives and efforts. They argue that extreme weather events are: " . . . not risks which are unknown . . . and it is not that the methods for coping do not exist . . . rather it is the inability of those most affected to cope to the impacts due to the lack of—or systematic alienation from accessing resources needed to guard against these events" [42] (p. 34).

Climate change is a complex phenomenon and generates multifaceted, distinct, and dynamic impacts in societies. These impacts cannot be addressed by using a silo approach, but rather, they require the application of myriad systems, approaches, and technologies. Jordan et al. [43], for example, are of the view that any single entity or actor is inadequately equipped to address and resolve challenges arising from changes in climatic conditions. This assertion, therefore, suggests that any attempts or efforts to formulate a response to the challenges of climate change require the adoption and implementation of holistic approaches, which are driven by a needs-based system, and which involve the participation of all stakeholders [44]. The unprecedented pace at which the world's climate is changing and impacting society requires that government institutions and instruments take a key posture of facilitating community initiatives and embracing all stakeholders in decision-making processes [5]. It is now obvious that self-ruling or self-regulated societal adaptation operating in silos is not sufficient, and governments must now play an active role in promoting collaborative systems in the search for solutions that bring about transformative adaptation to climate change [45,46]. This will further require significant changes in governance systems and processes; from the current bureaucratic ones to ones that ensure that government effectively interacts and builds effective, cooperative alliances with communities and the

private sector [47] (pp. 83–104). The success or failure of societies, communities, and households in adapting to the impacts of climate change is highly dependent on the nature and effectiveness of governance [38,48]. Filho et al. [29] argue that a key factor in enhancing climate adaptation is the strengthening of institutions and implementation of well-designed national and city-level planning policies and governance systems.

An effective and efficient governance system is a noble dashboard indicator for measuring the preparedness of any government entity to climate change challenges c. [49]. Chhotray and Stoker [50] and Richards and Smith [51] are of the view that a comprehensive and effective governance system which fosters the principles and values of stakeholder engagement and participation provides more opportunities for cultivating an inclusive platform for generating rich discussions and policy options. Gillard et al. [52] and Termeer et al. [53] further observe that, while centralised systems of governance can facilitate the coordination and prevent overlapping and duplication of programmes and the allocation of resources, they can also potentially prohibit experimental learning, trust-building, and collaborative management, and disregard local priorities and context sensitivities [54,55]. Thus, Armitage et al. [56] advocate for the breakdown of systems and processes which act as barriers for promoting the inclusive engagement and participation of various stakeholders, including grass-roots communities, in the identification of solutions to problems that affect them.

Binns et al. [20] and Cheru [41] argue that governments must move away from the naivety of believing that they are better placed to make key decisions on behalf of the poor people and to steer community development. On the contrary, they must pay attention to the critical responsibilities and energy that the poor in local communities expend in resolving the myriad socio-economic and environmental problems they encounter daily. Alemaw and Simatele [16] argue that a characteristic challenge to the adaptation agenda of poor people is not the scale of their socio-economic status, but the weakness of national and local government institutions and governance systems in the face of unprecedented impacts arising from climate and environmental changes. At all levels of government, a lack of resources and knowledge has tended to prevent people and institutions from solving problems and mapping change triggered by climate change [16]. This situation has further resulted in the absence of any comprehensive climate change adaptation intervention measures and has destabilised the productivity of the poor [57]. Mubaya and Mafongoya [58] observe that a lack of transparency and inclusivity in policy discussions almost always results in the marginalisation and the disenfranchisement of the poor and those with little or no political or financial power to influence the direction of any policy formulation.

Despite the frequent exclusion of the poor from participating in policy discussions, existing literature suggests that, if given the right impetus and empowered to define their development, they can resolve many of the challenges they face [57]. Hsieh and Lee [59] argue that cultivating the participation of local people in decision-making always has the benefit of stimulating the achievement of several important objectives, among which are fostering a greater sense of commitment, involvement, and ownership of the resolutions reached; and second, delivering adaptation services which are much needed by the community. Edifying this observation, Hove et al. [60] are of the view that there is an urgent need to shift the paradigm of policy engagement from a top-down narrative to one which promotes and is based on a strong and genuine grass-roots-grown and -driven process. Simatele et al. [38], on the other hand, argue that a people-centred approach, through community involvement, can create sensitivity to, and enhance the nature of, the adaptation strategies which the poor can use to build their adaptive capacity and resilience to climate-induced, environmental changes. Zeidler et al. [61] further observe that any national strategy for building the resilience of the poor to climate change must be conducive and aligned to local conditions and the coping mechanisms or adaptation practices employed by the local people, which are usually rooted in indigenous knowledge systems (IKS) and community-based innovation. Kettle et al. [62] and Coffey and O'Toole [63] therefore advocate the urgency of obtaining governmental and non-governmental actors to

embrace traditional practices, which are ingrained in IKS, in their advisory services so as to formulate robust adaptation options for the targeted audiences.

A major challenge to the formulation of transformative climate change adaptation lies in what Leck and Simon [64] have deemed the "siloed and hierarchical approach" to climate change governance. They argue that, historically, this approach tends to negatively counter the connectedness and collaboration required for the development of more transformative systems and processes for community adaptation. The multidimensional challenges associated with climate change necessitate the need for collaboration and partnerships across multiple sectors, scales, and actors, so as to address climate change problems [16]. Thus, a more integrated approach to adaptation offers better opportunities to formulate comprehensive systems of climate change coordination and response vis-à-vis adaptation action [65–67]. Improved collaboration between knowledge generators, intermediary governmental and non-governmental agencies, and end-users is a key requirement in capturing the unique contexts in which we can come to understand the impacts of climate change [68–70]. Improving collaboration between governmental agencies and other stakeholders, including local communities, can aid in the effectiveness and efficiency of adapting knowledge delivery to end-users [65]. Evidence suggests that facilitating localised collaborations between municipalities and higher levels of government contributes to extending interventions beyond one geographical scale, as well as to integration across policy scales [16,64]. The potential for multi-stakeholder partnerships to adopt a flexible, decentralised, and inclusive structure [71,72] appeals—theoretically at least—to the idea that adaptation should be implemented locally, where vulnerability is experienced [73]. A combination of leadership, local government support, and stakeholder buy-in has been proven to be necessary to implement adaptation measures which meet the aspirations of the communities most affected by climate change [74,75].

Kalafatis et al. [76] and Ochieng [77] further argue that improving communication can significantly enhance knowledge uptake. A considerable body of literature from developing and developed countries emphasises the need for reliable and comprehensive climate change communication to influence the sharing and shaping of knowledge in adaptation and decision-making [78–86]. Kvamsås [87] observes that, even though the growing body of scientific knowledge does not lead to growing consistency in societal attention, political commitment, or state interventions [88], knowledge connects climate adaptation and local political agendas, influencing priorities and anchoring decisions. Undoubtedly, there is still much to be done to improve the supply of robust knowledge to policymakers and practitioners to enhance communities' capacities to adapt to climate change. Institutional capacity is the strongest predictor of national adaptation policies and action [89]. Evidence suggests that an effective governance system which cuts across sectoral and multilevel endeavours is vital for the efficient coordination of different climate change actors, and that development is a comprehensive and forward-looking adaptation agenda [90,91]. Mapfumo et al. [92] highlight the importance of political will and political feasibility to undertake coordinated measures of a transformational nature in response to the threats of severe climate impacts. In conclusion, therefore, it can be argued that leaders at all levels of society play a role as brokers, often as the glue that brings together different societal actors, thus enabling not only committed and effective community participation, but also assigning to that participation a meaningful place in governance [93,94]. Thus, the development of any climate change adaptation intervention measures and strategies requires the setting into motion of several considerations, among which the inclusive engagement and participation of different stakeholders and communities are paramount.

## 3. Materials and Methods

The primary data on which this paper is based was collected between February and October 2021 in the KwaZulu-Natal Province of South Africa. (See Figure 1). The study employed a combination of participatory research tools, including online surveys and semi-structured interviews, to explore the perception of communities on issues re-

lating to climate change adaptation governance in South Africa. A purposive sampling method was used to identify the key stakeholders in the four local municipalities of the uMkhanyakude District Municipality in northern KwaZulu-Natal: Big 5 Hlabisa Local Municipality (*n* = 27), uMhlabuyalingana Local Municipality (*n* = 28), Mtubatuba Local Municipality (*n* = 42), and Jozini Local Municipality (*n* = 29). These key stakeholders were drawn from different sectors of the community and included councillors, religious/church leaders, iziNduna (tribal councillor(s) or headmen among the Zulu people of South Africa), Amakhosi (chiefs or leaders of communities or tribes among the Zulu people), representatives from different non-governmental organisations, community leaders representing different community-based organisations, traditional leaders, business leaders, and school principals. Due to the challenges of COVID-19, the questionnaires were distributed online using the Google Forms platform. The online survey consisted of multiple-choice questions, dropdown questions, and open-ended questions. This method of data collection was highly suitable for a multifaceted topic such as climate change as it enabled the researchers to explore rich context-oriented discussions grounded in the lived experience of the key stakeholders, e.g., [95].

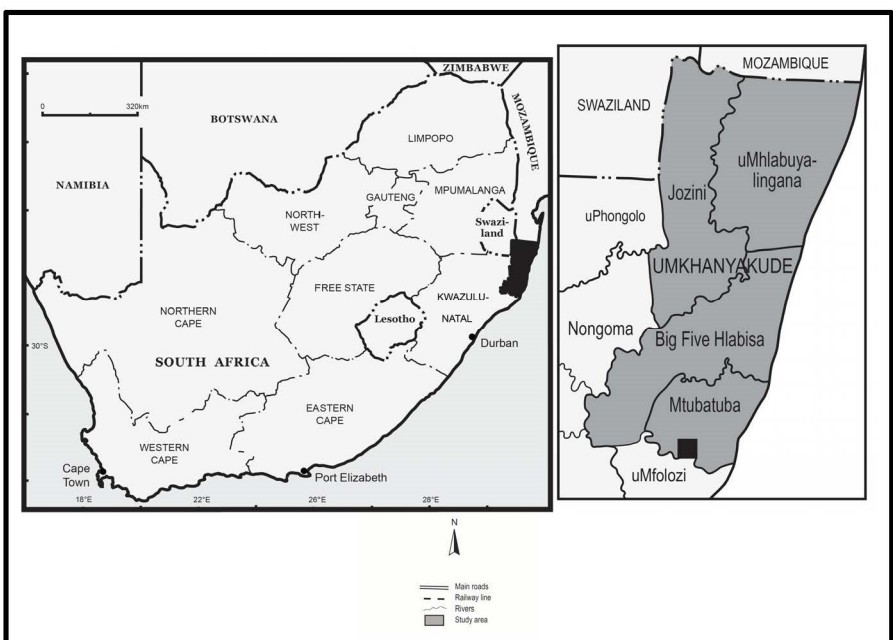

**Figure 1.** Location of the study sites in KwaZulu-Natal Province, South Africa. **Source:** Cartography Unit, University of the Witwatersrand (2021).

Additionally, the study used semi-structured interviews with officials from the Department of Forestry, Fisheries and the Environment (DFFE) (*n* = 6), KZN Department of Economic Development, Tourism, and Environmental Affairs (EDTEA) (*n* = 3), and local municipalities (*n* = 4). The semi-structured interviews were conducted telephonically and online through Microsoft Teams. The officials from government departments provided expert viewpoints related to climate change governance in South Africa. While the use of a limited selection of interviewees has the potential to draw criticism, many scholars have emphasised the wealth of understanding that can come from a narrow pool of perspectives, as discussed in [96].

The data obtained through online surveys were exported from Google Forms to a Microsoft Excel spreadsheet, where they were subsequently cleaned, coded, and analysed. Furthermore, the qualitative data were quantified using descriptive statistics (i.e., frequencies). In contrast, the data obtained through semi-structured interviews were digitally recorded and professionally transcribed. Major themes emerging from the qualitative data were compared and then organised to create more concise narratives that would

unify the data within each category. Both the quantitative and qualitative data enabled the researchers to analyse the myriad ways in which climate change adaptation policies and strategies are developed, as well as the factors which facilitate involvement in the discourse.

## 4. Results—Climate Change Governance and Community Adaptation in KwaZulu-Natal Province

Although climate change is a global concern, its effects are contextually specific and vary across countries, regions, and communities. Understanding these different impacts, therefore, requires that they are spatially contextualised. In view of this assertion, one of the key points of investigation for the study on which this paper is based involved the assessment of the research participants' understandings of climate change and its impacts on the community. To achieve this, a question requiring research participants to indicate their knowledge of climate change was asked, and their responses are presented in Table 1.

**Table 1.** Participants' Knowledge of climate change and impacts in uMkhanyakude District Municipality.

| Type of Response | Knowledge of Existence of Climate Change | (%) | Knowledge of Climate Change Impacts on Community | (%) |
|---|---|---|---|---|
| Yes | 120 | 95.2 | 114 | 90.4 |
| Maybe | 5 | 4 | 3 | 2.4 |
| I don't know | 1 | 0.8 | 6 | 4.8 |
| No | 0 | 0 | 3 | 2.4 |
| Total number | 126 | 100 | 126 | 100 |

**Source:** Field-based surveys (2021).

Scrutinizing Table 1 suggests that an estimated 95% of the research participants expressed knowledge of climate change, with 90% claiming that they have knowledge of the impacts of climate change on their communities. Only an estimated 0.8% of all of the respondents indicated not having any knowledge of climate change, with 2% indicating a lack of awareness of the climate change impacts on their community.

It was also imperative to have an understanding of what community members consider to be the manifestation of climate change in their respective communities. They were thus asked to identify any weather event(s) they ascribed to being a manifestation of changes in climatic conditions (c. Table 2).

**Table 2.** Observed impact of climate extreme events in uMkhanyakude District Municipality.

| Impact of Extreme Climate Event | Cumulative No. Responses | (%) |
|---|---|---|
| Declining water supply | 118 | 19.6 |
| Reduced agricultural yields | 109 | 18.1 |
| Increased livestock losses | 105 | 17.4 |
| Health implications | 100 | 16.6 |
| Reduced work and business opportunities | 86 | 14.3 |
| Deterioration in socio-economic status | 84 | 13.9 |
| Total number | 602 | 100 |

**Source:** Field-based surveys (2021).

The information in Table 2 suggests that an estimated 20% of all of the responses alluded to the declining water supply in the region as a factor of climate change, while 13% of the respondents pointed to deterioration in socio-economic status as a direct result

of climate change. What is important to note is the combined impacts on livelihood-supporting systems, which is estimated to be 55% of the total responses.

Although it can be argued that the above assertions are perceptions and lack scientific basis, they are indications that changes in climatic conditions are presenting a number of challenges for local and rural communities in South Africa. It was of paramount importance to establish community perceptions on organisations they consider important in facilitating their adaptive capacity against the impacts of climate change. Figure 2 is an illustration of these perceptions.

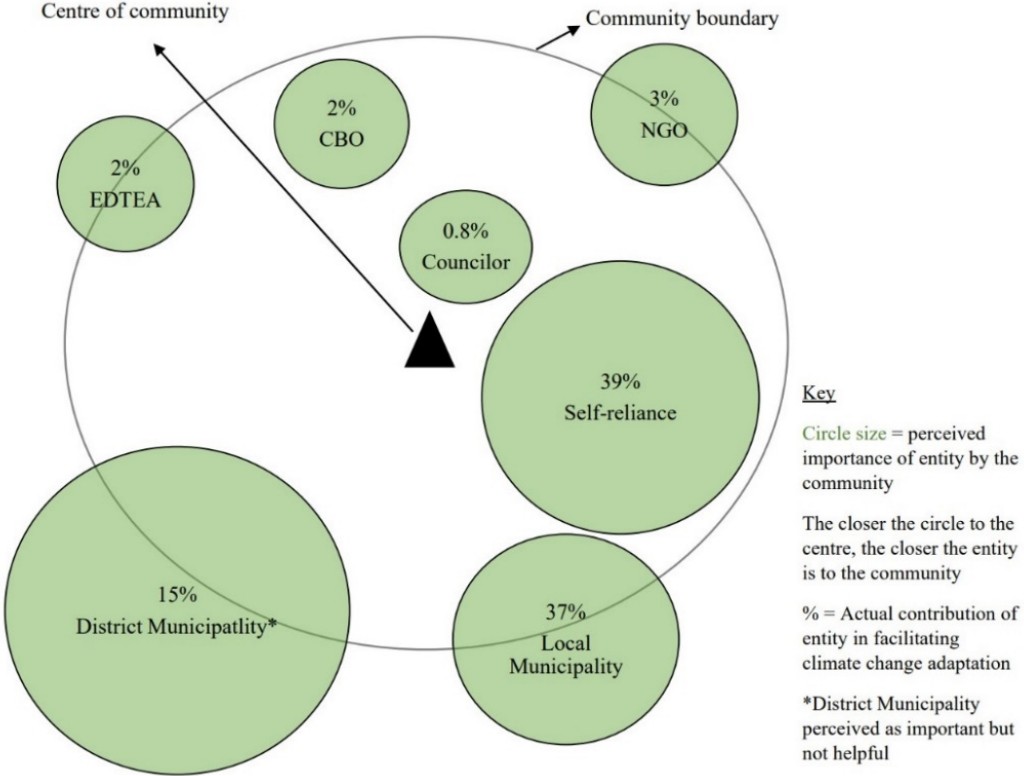

**Figure 2.** Community perception of important organisations in facilitating climate change adaptation in the study sites. **Source**: Field-based survey notes (2021).

An examination of Figure 2 suggests that, although the district municipality is considered by community members as the most important official entity (i.e., as judged by the size of the circle), which ideally should facilitate the adaptive capacity to climate change, it is far removed from the centre of the community, and it only provides about 15% of the assistance provided in responding to climate change challenges. Many community members (i.e., 39%) seemed to suggest that dealing with climate change challenges rests entirely on the community itself. However, closer scrutiny of Figure 2 suggests that, although the entities identified by the community are represented as being far removed from the centre of the community, they provide a combined 59.8% of the assistance for community responses to climate change.

Another important component for this study was the need to evaluate the effectiveness of any of the climate change outreach programmes and activities used by climate change practitioners in facilitating community adaptation to climate change challenges. Research participants were thus asked to rate the level of effectiveness of the activities. These views are illustrated in Table 3.

**Table 3.** Level of effectiveness of climate change initiatives, policies, and strategies in uMkhanyakude District Municipality.

| Level of Effectiveness of Climate Change Activities | No. Responses | (%) |
| --- | --- | --- |
| Not Effective | 93 | 73.9 |
| Less Effective | 24 | 19.0 |
| Very Effective | 9 | 7.1 |
| Total number | 126 | 100 |

**Source:** Field-Based survey (2021).

Table 3 reveals that 73% of the responses suggest that the existing climate change initiatives and strategies are not effective, while only 7% see them as effective. In view of this, it was imperative to ascertain the level of community awareness of any existing climate change initiatives and strategies used to promote their resilience and adaptation to climate change. Research participants were asked to rate the level of effectiveness, by way of indicating whether they were aware of any existing climate change initiatives, policies, and/or strategies for community adaptation. Figure 3 illustrates these responses. Figure 3 suggests that an estimated 60% of the responses indicated a lack of knowledge of any climate change initiatives, policies, and strategies for community adaption to climate change. A further 19% indicated knowledge of climate change initiatives for community adaptation, while 21% expressed a view of being moderately aware. What is of particular interest in Figure 3 is that an estimated 36% of the 60% of the responses expressing a lack of knowledge of any climate change initiatives and/or policies were women, while only 7% of the total 19% claiming to have knowledge of climate change policies, strategies, and activities for community adaptation were women. The overall impression represented in Figure 3 is indicative of the fact that women seem to be the group of people who are generally unaware of any community initiatives, policies, and strategies for adaptation to climate change. On the one hand, Figure 4 suggests that an estimated 88% of all of the responses from the research participants indicated neither participation nor involvement in climate change activities, strategies, or policies for community adaptation, with only 7% indicating participation in community climate change adaptation discourses.

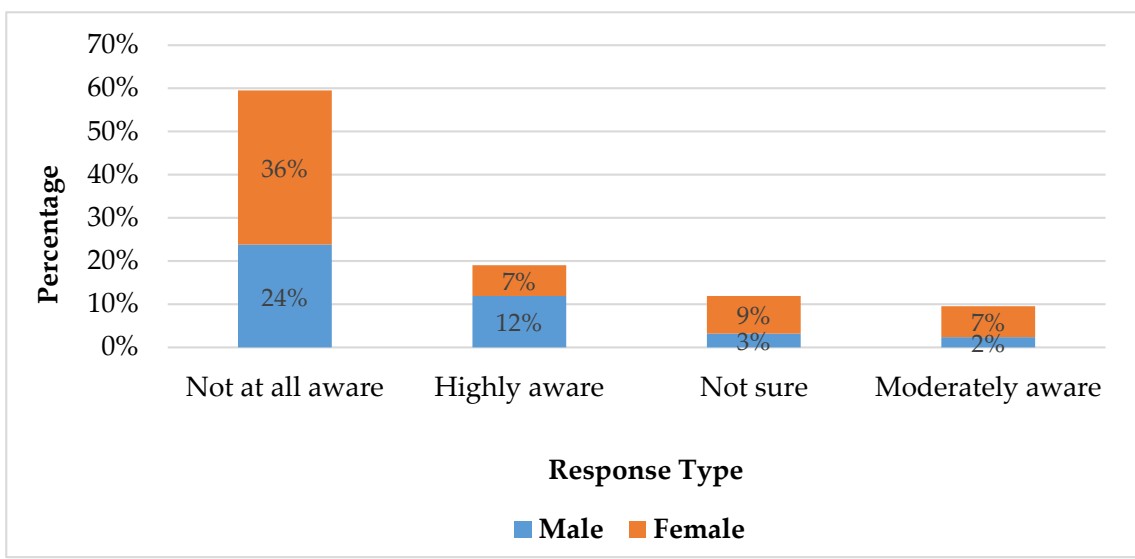

**Figure 3.** Level of community awareness of climate change initiatives, policies, and strategies in uMkhanyakude District Municipality. **Source:** Field-based surveys (2021).

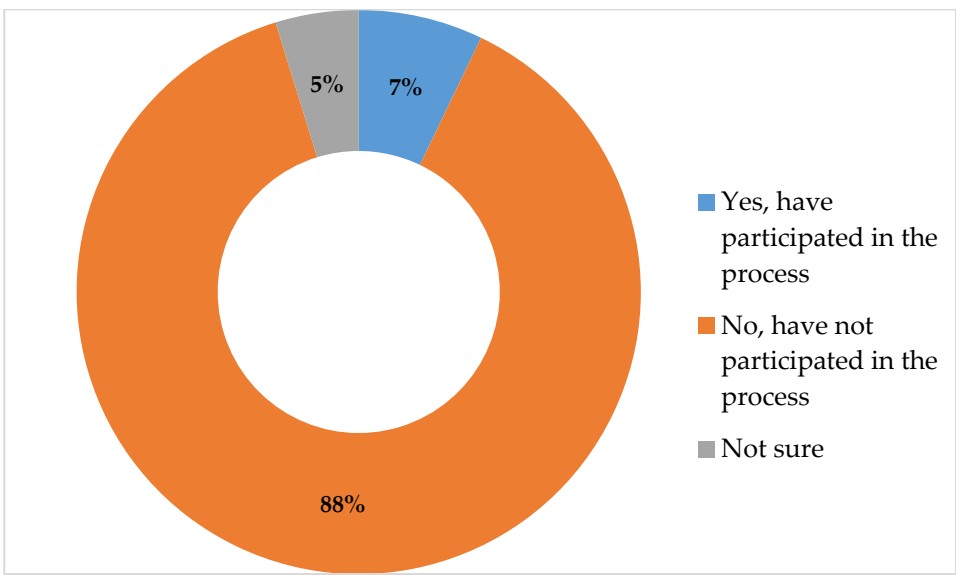

**Figure 4.** Level of participation by community members in climate change initiatives, policies, and strategies in uMkhanyakude District Municipality. **Source:** Field-based surveys (2021).

In view of the above assertions, it was imperative to have an understanding of the climate change adaptation policy landscape in South Africa in terms of its formulation. This was in a bid to comprehensively contextualise the opportunities and barriers that may exist in facilitating or hindering community participation in climate change discussions at the grass-roots level. Thus, discussions with policy practitioners drawn from different sectors of the government, quasi-organisations, and NGOs in uMkhanyakude District Municipality were conducted. These technocrats were engaged in articulating the policy-formulation processes, with particular interest paid to understanding the level of stakeholder participation and how this element is espoused and embedded in the entire policy-formulation framework. Figure 5 is a depiction of the policy-making process for climate change adaptation in South Africa.

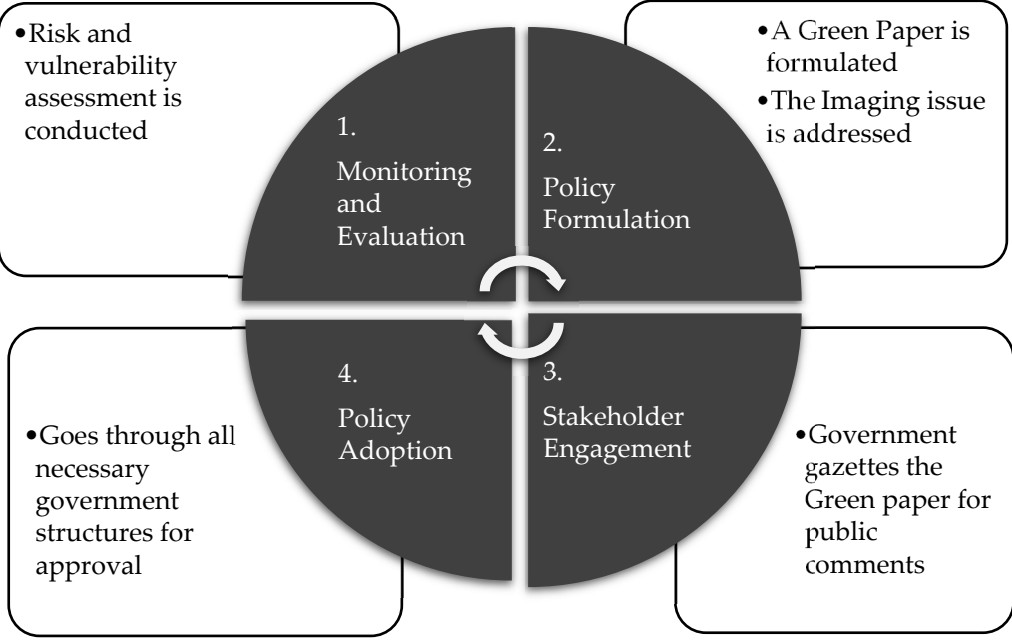

**Figure 5.** The South African climate change adaptation policy-making cycle. **Source:** Field-based interview notes (2021).

Policy formulation for climate change adaptation in South Africa goes through a four-stage process, as illustrated in Figure 5. It is issue-based and driven by identified risks arising from climate change.

As a result of the comprehensive policy-formulation processes as illustrated in the above policy-making cycle, South Africa has developed a number of policy instruments and strategies for promoting community adaptation. It was thus important to identify the existing instruments which are used to promote climate change adaptation among rural communities in South Africa. Figure 6 depicts some of the instruments which guide the climate change adaptation agenda in the uMkhanyakude District Municipality.

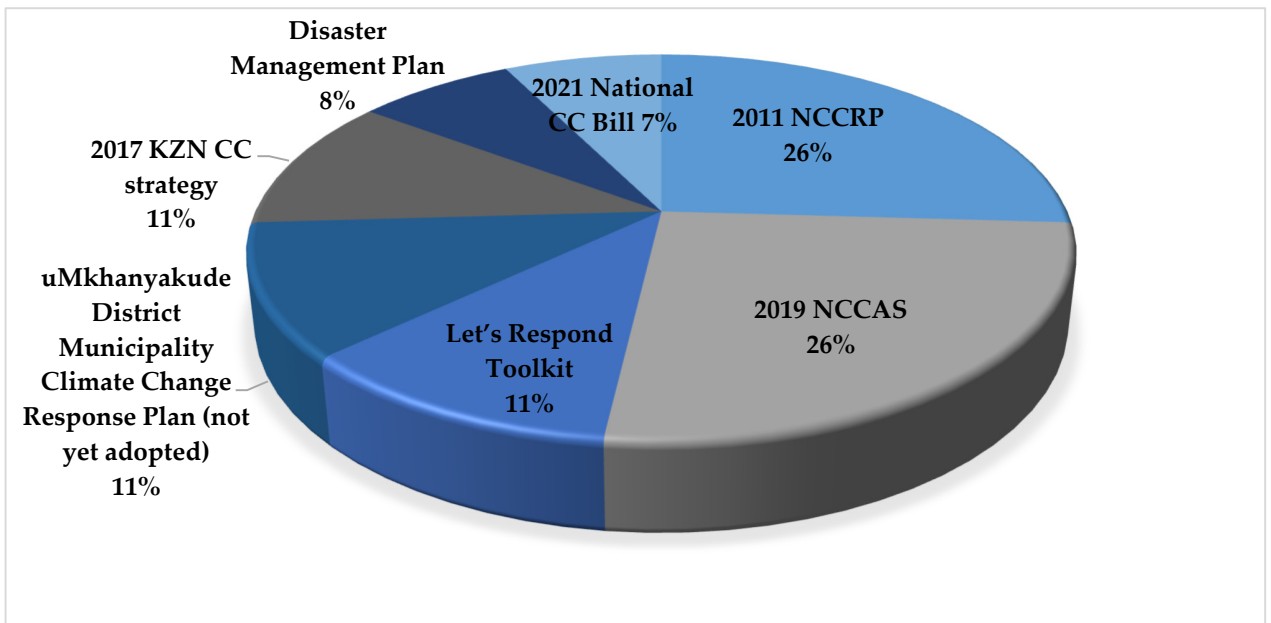

**Figure 6.** Climate change instruments for climate change adaptation in the study sites. **Source:** Field-based interview notes (2021).

An important element emerging from Figure 6 is that the National Climate Change Response Policy (NCCRP) and the National Climate Change Adaptation Strategy (NC-CAS) are two policy documents which predominantly guide the formulation of climate change adaptation strategies at the national and local/or community levels. In view of the strides made in developing a climate change adaptation framework in South Africa, it was necessary to establish the extent to which these national policy frameworks influence the formulation of comprehensive strategies and initiatives for community adaptation to climate change. Officials were thus asked to indicate whether, as technocrats, they perceived these policy instruments and initiatives to be effective in facilitating the adaptive capacity and resilience of grass-roots communities to climate change. Table 4 reveals that all (i.e., 100%) officials indicated that climate change initiatives, policies, and strategies have great potential in influencing community adaptation in a positive way.

**Table 4.** Perceptions of officials on the effectiveness of adaptation instruments to climate change in the study sites.

| Type of Response | No. Cumulative Response | % |
|---|---|---|
| Yes, instruments are effective for adaptation | 13 | 100 |
| No, I don't think so | 0 | 0 |
| I am not sure, maybe they do | 0 | 0 |
| Total number | 13 | 100 |

**Source**: Field-based interview notes (2021).

From one point of view, it can be argued that, although South Africa has developed good climate change initiatives, policies, and strategies, the implementation of these policies seems to present difficulties as those for whom they have been developed do not seem to have any knowledge of their effectiveness in helping them build resilience against extreme weather events. In view of this assertion, it was important to have some level of understanding of factors which may contribute to making these intervention measures less effective. Table 5 depicts the views of research participants on what they consider to be major barriers to the development of an effective climate change adaptation agenda in the study sites.

**Table 5.** Barriers to comprehensive climate change adaptation in the study sites.

| Perceived Barriers | Qualitative Examples from Research Participants | Solutions: Research Participants' Views |
|---|---|---|
| **Lack of informative interactions with the communities** (e.g., awareness campaigns) **Lack of local expertise** (e.g., no dedicated climate change practitioners within the district and local municipalities) **Competing political priorities Cognitive and organisational silos Lack of public engagement** | "I feel that the municipality has told us that the weather or climate change is not a priority for them". "District or UMkhanyakude never conducts formal workshops for councillors; they also do not call public meetings to facilitate climate change workshops". "Nothing is done by the Municipality. They also do not educate nor inform us of the existing climate change adaptation plans. The Municipality never asks for the communities views. In short, I don't think the Municipality knows how to best handle climate change governance". "I don't see local authorities doing much about climate change". "I think it is not effective because the communities we are living in are poor but the Municipality wants payments for any service rendered to the communities, where are these people supposed to get the money as I have already said most of them are unemployed". "If it was effective then the district would not be in such dire state. Plans need not only talk to climate adaptation but also touch on mitigation to ensure a proactive approach and not a reactive one. Currently, climate change is viewed as a disaster impact at uMkhanyakude, which means the municipality reacts to the impacts not necessarily plan by putting measures in place i.e., air quality monitoring or early warning systems, etc. I am not sure of the level of mobilisation or infiltration into other business units or sectors, or whether the strategy remains a strategy for only Disaster management and environmental management officials. Not sure whether other sectors, i.e., water, human settlements, planners understand their role in managing climate change. Furthermore, the plans need to be budgeted for and much effort needs to also go towards involving other non-governmental/community-based organisation to ensure holistic regional implementation". "In most cases when these policies are developed community people are not contacted, therefore the most important information is left out". "After being affected by drought no one was able to address our loss, not even by conducting awareness campaigns on extreme weather events and strategies to cope". "No public engagement or participation was conducted when developing the strategies, I feel like the communities are left out when such strategies are developed and it becomes difficult for the Municipalities to implement the climate change adaptation strategies". | "Conduct climate change awareness campaigns—Educate the communities and encourage them to advocate for climate change issues". "Active participation and involvement—be inclusive". "Bottom-up Approach include communities views and engage with them. The municipality must have meetings with community members and also take suggestions from them". "Building collaborative partnerships between government, NGOs, CBOs, municipalities, traditional structures and communities". "Enhance youth participation". "Include all sectors within uMkhanyakude District Municipality and develop skills related to tackling climate change". "Building capacity—train people within the communities who can teach and engage with the communities on climate change issues". "Transparency in policy development and implementation is important". |

**Source**: Field-based survey notes (2021).

A key theme emerging from Table 5 revolves around the lack of community engagement, collaboration, and transparency among the different actors mandated to manage issues relating to climate change and socio-economic development. It is evident from the views articulated by the participants that there are no comprehensive interactions and engagements between the uMkhanyakude District Municipality, the community, and

other government departments who, in earnest, should work together towards developing integrated adaptation policies and strategies. The silo approach to dealing with climate change has created a situation where many of the rural people in the study sites have learnt to be self-reliant when responding to climate-change-induced challenges.

## 5. Discussion

The empirical evidence presented in this paper suggests that climate change is having devastating impacts on the livelihood-supporting systems of many poor and rural households in South Africa. These changes are not only affecting the biophysical elements of the natural resource base, but they are also greatly compromising the ability of many small-scale farmers, especially women, who, in most cases, lack any form of productive assets to deploy so as to minimise the impacts of climate change. These findings are not unique to South Africa, but rather they seem to align with the findings of other studies in the sub-region. Simatele et al. [38], for example, argue that the changes in climatic conditions are increasingly limiting the ability of the poor to actively engage in meaningful agricultural activities, owing to the eroding effects of climate change on their asset portfolios. This view is supported by Diffenbaugh and Burke [97] (p. 9880), who observe that "there is growing evidence that poorer countries or individuals are more negatively affected by a changing climate, either because they lack the resources for climate protection or because they tend to reside in warmer regions where additional warming would be detrimental to both productivity and health. The increase in vulnerability of the poor to climate change requires an urgent and concerted policy response to their plight.

There is, therefore, an urgent need for government systems and policies to go beyond the syndrome of simply acknowledging the impacts of climate, but rather moving in a space which ensures and enables human and natural systems to adjust to the actual and projected climate stimuli and impacts. Thus, the formulation of pro-poor policies and strategies which promote gender equality in issues of climate change must be prioritised as they have the potential to facilitate the identification of appropriate intervention measures which can contribute to building the adaptive capacity and resilience of the most vulnerable in society. Demetriades and Esplen [98] (p. 27), for example, argue that " . . . achieving a gender balance in participation in climate change negotiations and representation at decision-making (vis-à-vis policy) tables is a good starting point . . . to successful (climate change) interventions". This view is supported by Villagrasa [99] (p. 41), as cited by Dennison [100], who argues that "promoting women's and girls' meaningful participation in decision-making also contributes to addressing gender inequalities by raising the profile and status of women and girls in the community and challenging traditional assumptions about their capabilities". Achieving this requires a radical shift towards designing gender-sensitive climate change responses and policies which incorporate the voices of the poor and powerless, as well as taking into consideration their local practices, including their indigenous knowledge.

A key element to an inclusive climate change adaptation policy which is transformative in nature and builds the resilience of the poor to current and future climate risks lies in the quality of its leaders and the sharpness of the country's governance systems in dealing with complex issues, as discussed in [41]. If South Africa is to comprehensively respond to the challenges of climate change, it will require the theoretical sharpness and practical abilities of both state and local authorities to adapt formal institutions to new and changing, climate-induced realities. As observed by Roelich and Giesekam [101], priority must be given to: (a) raising climate change awareness and agricultural skill development among rural and vulnerable groups of people in order to improve productivity and reduce poverty, unemployment, and helplessness; (b) strengthen the level of community participation in policy discussions and formulations. Such an approach would result in the development of optimal intervention strategies which are based on the lived realities of those most impacted by climate change; (c) promoting equity through the opening up of the political process

to poor and powerless people to influence economic efficiency and, more specifically, the control of climate-change-related financial resources.

It is not an exaggeration to argue that the rural poor of South Africa are expending tremendous dynamism and vivacity to change their fortunes in the face of changes in climatic conditions. All they require is a supportive framework through which they can pursue their ambitions and aspirations to build their adaptive capacity and resilience to climate change. Moser and Satterthwaite [39], for example, argue that the poor are not passive actors, but rather they are actively involved in protecting and modifying their asset portfolios against the impacts of climate change. This view is supported by Nicoson and von Uexkull [102], who argue that, while climate change does not cause weak governance, weak governance limits individuals' and communities' coping capacities in the face of climate change. We can thus argue that, while state and local authorities have failed to adequately articulate new visions or provide necessary services for climate change adaptation, there is an urgent need to mobilise community resources and citizens' groups to charter an adaptation agenda that is suited and responsive to the needs of those most vulnerable to climate change.

As argued above, it is not the incapability of the poor to adjust to changes in climatic conditions that worsens their vulnerability, but rather it is the poor governance systems and weak policies and strategies which harm the everyday lives of the poor. In as much as South Africa has some of the best social policies on the African continent, the country has for too long lacked good governance, an essential ingredient in the development of transformative climate change adaptation policies and strategies. The prevailing approach to governance, based on a central government, places very little emphasis on the importance of private agents and actors or citizens' groups in devising locally tailored, innovative approaches for climate change adaptation. It allows the vested interests of politically and economically powerful elites to influence unduly the nature of policies and strategies which are then implemented by local authorities and which are devoid of the lived experiences of the poor. Flato et al. [103] and Cheru [41], for example, argue that African governments must create a climate of equity, cooperation, and accountability, in which the talents of all their citizens can be applied to solving the climate change problems they face. This will require significant reforms in governance systems in terms of the representativeness of society in decision-making processes. The engagement of civil society in fair and transparent policymaking, promoting ethos in partnerships with local authorities, community-based organisations (CBOs), traditional leaders, and the private sector, can make the difference between formulating a good, pro-poor climate change adaptation agenda, or one which has misplaced priorities.

In conclusion, therefore, the importance of co-designing climate change policies and strategies with communities cannot be overemphasised, particularly because the communities represent the voices of the most vulnerable and marginalised groups in South Africa. Thus, in order to achieve successful public participation in climate change adaptation policy development, there must be an all-inclusive system which incorporates all stakeholders, including vulnerable groups. Bahauddin et al. [104] advocate strong relationships and collaboration between the government and different stakeholders if comprehensive climate change adaptation policies are to be developed. Thus, there is an urgent need for South Africa as a country to establish working institutions that are invested in fulfilling their mandates and supporting communities in building strong partnerships between civil society, public–public, and public–private sectors. Such partnerships may trigger the achievement of effective and transformative systems of climate change governance and building the resilience of the poor. The United Nations [105] has observed that "business as usual" approaches will not in any way achieve climate-resilient development for the world's most vulnerable people. Adger et al. [48] and Adger et al. [106] further argue that the success or failure of climate change adaptation is highly dependent on the effectiveness of governance, whilst Filho et al. [29] (p. 35) are of the view that the key factors to enhance climate adaptation are based on "strengthening institutions, designing well formulated

national and city-level planning policies for systematic governance of social, economic and political processes which are essential for navigating the climate crisis". If South Africa is to adapt effectively to climate change, the case for institutional reformation and strengthening in ways that embrace the voices of the marginalised groups, is therefore, an important ingredient. In the absence of these elements, transformative adaptation to climate change will remain a myth.

## 6. Conclusions, Limitations and Recommendations

In this paper, it has been established that climate change in South Africa, like any other African country, is having devastating impacts on social and economic aspects. The poor and rural households that are mainly dependent on agriculture productivity for their livelihoods and income generation are the most affected. Furthermore, it has been demonstrated that any attempts to grasp the complexity and dimensions of climate change require flexibility and a keen interest on the part of state and local authorities to intensify support for those most impacted by it. Given the interrelated nature of climate change on social and economic processes, state and local authorities, including their cooperating partners, must move away from narrowly focused, sectoral perspectives and towards more inclusive, multidisciplinary and participatory approaches. The extent to which transformative adaptation to climate change will be achieved will depend on the extent and appropriateness of innovative approaches to climate governance. There is a need for South Africa to base its adaptation agenda on the lived experiences of the poor, and the strategies to be developed must reflect the local needs for climate change response. Thus, the empirical evidence presented in this study should be used as an entry point through which the government can enhance preparedness plans for climate change adaptation, particularly for vulnerable groups who oftentimes lack the voice and power to influence policy direction. This will also contribute to UN Sustainable Goal 13, which speaks to strengthening resilience and adaptive capacity against climate-related hazards and natural disasters.

This research, however, is subject to several limitations. First, this study focused on the perceptions of key individuals in the community. Although engaging with key individuals in the community is important in acquiring a better understanding of the local community needs and resources, household members are the ones who navigate climate change issues on a daily basis. Thus, future studies could investigate the individual household members' views and perceptions on climate change governance in South Africa.

Secondly, the study used online data collection instruments. This method posed a limitation because digitally illiterate participants could not participate in the survey. Thus, this led to a digital divide, which increased the social exclusion of already-vulnerable groups. Future studies must ensure that the data collection instrument used allows for equal and fair opportunities for all groups and individuals to participate in the study.

**Author Contributions:** Conceptualization, N.S. and M.D.S.; methodology, N.S. and M.D.S.; validation, N.S., M.S., L.M. and M.D.S.; formal analysis, N.S.; investigation, N.S.; writing—original draft preparation, N.S.; writing—review and editing, N.S. and M.D.S.; supervision, M.D.S.; funding acquisition, M.D.S.; resources, M.S. and L.M. All authors have read and agreed to the published version of the manuscript.

**Funding:** This research was funded by the National Research Foundation of South Africa, Research Grant No. 129481; Ref RCUZ200513521731 and the University of the Witwatersrand Postgraduate Merit Award.

**Institutional Review Board Statement:** The study was reviewed and approved by the University of the Witwatersrand Ethics Committee (protocol number H20/10/28, date of approval: 16 October 2020).

**Informed Consent Statement:** Written informed consent has been obtained from the participants to publish this paper.

**Data Availability Statement:** The original contributions presented in the study are included in the article; further inquiries can be directed to the corresponding author.

**Acknowledgments:** This paper and the research behind it would not have been possible without the exceptional support of my supervisor, Mulala Danny Simatele from the Global Change Institute (GCI) at the University of the Witwatersrand. I am also grateful for the support of my family, friends, and all of the research participants who gave me their time.

**Conflicts of Interest:** The authors declare no conflict of interest. The funders had no role in the design of the study; in the collection, analyses, or interpretation of data; in the writing of the manuscript; or in the decision to publish the results.

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
