# Peer review of "Empowering the Voiceless: Securing the Participation of Marginalised Groups in Climate Change Governance in South Africa"

_sustainability, doi:10.3390/su14127111_

Round 1

Reviewer 1 Report

This paper investigates the community perception of the climate change adaptation governance in South Africa by assessing the developed climate change adaptation policies and strategies and the related parameters using a survey of all involved stakeholders in the development of these intervention measures.

The research is rather interesting and well-organized but needs some minor revisions. The main concern I have is that the result section is not well written in a scientific manner. For example, please see L 276-284, L 342-350, L 351-359, L 376-380, L 382-386. They are not similar to a scientific statement and should not be seen in a research paper. It is mostly a report. The authors should not mention some personal attitudes in a research paper.

Second, the conclusion section is too short and does not discuss the application of the main findings of this manuscript. The authors should explain how the main findings and contributions of this paper can improve the community perception or how they can help to enhance the preparedness plans.

Additionally, the conclusion section does not include any discussion or recommendation regarding future studies.

Author Response

Thank you for your contructive feedback and suggestions.  It is your constructive feedback and insightful comments that led to possible improvements in the current version.

Reviewer 2 Report

Dear Authors. 

This is an interesting paper addressing the vulnerabilities of marginalised groups of people in the selected study area in the KwaZulu-Natal portion of South Africa,  who many times are without the voice and power to influence policy direction on climate adaptation and mitigation.  The literature contextualisation is good and very meaningful results have been generated. However,   the manuscript has a few problem areas that are easily resolvable to give it  a more cohesive and balanced overview. 

  1. The authors do not spell out a clear research problem although they have succeeded in providing the background rationale and justification for the study. They are requested to use scientific action verbs such as to 'evaluate' or 'assess' or 'critique' etc so that the readers know directly what is the purpose of the research?
  2. In their sampling framework, its indicated that online interviews were conducted which is an acceptable study design. However, its not clear on to what extent were other potential respondents excluded, especially those who are not digitally literate? Alternatively, the authors can state clearly either in the method section or conclusion sections how this posed as a limitation. 
  3. The conclusion and recommendation given are very brief or rather too short and sketchy relative to the extensive results that were presented and discussed. Can you please  link and integrate a little with the paper background and enrich your synthesis in the conclusions mentioned. 
  4. Will you please indicate recommended future research directions or fruitful areas for future research in the relevant section. 
  5. Lastly, please read your Abstract again. It is heavy on the study rationale and motivation but thinner on the research problem and the result highlights  and  the conclusions that came out of this study. 

I enjoyed reading your manuscript and the commentary given are offered in the spirit of improving and closing loopholes and giving a balanced view. 

Best wishes with the revisions, 

Sincerely,

Independent reviewer. 

Author Response

Thank you for your contructive feedback and suggestions. It was your constructive feedback and insightful comments that led to possible improvements in the current version. 

Reviewer 3 Report

Line 288: Change illustrating to illustration